Stage 1 registered report: spatiotemporal patterns of the COVID-19 epidemic in Mexico at the municipality level

http://orcid.org/0000-0002-6138-9879 Mas Jean-François jfmas@ciga.unam.mx
Laboratorio de Análisis Espacial, Centro de Investigaciones en Geografía Ambiental, Universidad Nacional Autónoma de México , Morelia, Michoacán , Mexico
Aly Sharif
Electronic publication date: 2021 Feb 5
Publication date: 2021
Volume: 9
Electronic Location ID: e10622
Received 2020 Aug 21; Accepted 2020 Nov 30
Copyright: © 2021 Mas
Copyright year: 2021
Copyright holder: Mas
License: This is an open access article distributed under the terms of the Creative Commons Attribution License, which permits unrestricted use, distribution, reproduction and adaptation in any medium and for any purpose provided that it is properly attributed. For attribution, the original author(s), title, publication source (PeerJ) and either DOI or URL of the article must be cited.
License URL: https://creativecommons.org/licenses/by/4.0/

Keywords: SARS-CoV-2 pandemic, COVID-19, Spatial analysis, GIS, Clusters, Spatial patterns, Spatial modeling, Spatial autocorrelation

Funding: The author received no funding for this work.

==============================
In this stage 1 registered report, we propose an analysis of the spatio-temporal patterns of the COVID-19 epidemic in Mexico using the georeferenced confirmed cases aggregated at the municipality level. We will compute weekly Moran index to assess spatial autocorrelation over time and identify clusters of the disease using the “flexibly shaped spatial scan” approach. Finally, different distance models will be compared to select the best suited to predict inter-municipality contagion. This study will help us understand the spread of the epidemic over the Mexican territory and give insights to model and predict the epidemic behavior.

Introduction

At this date (October 15), the most affected WHO region by COVID-19 pandemic, with 18,090,384 confirmed cases, is the Americas. In this region, the countries that present the most substantial number of deaths are the USA, Brazil and, Mexico.

According to the Mexican Health Secretary’s records, the first confirmed cases of COVID-19 in Mexico were reported in February 2020, and community transmission started at the end of March. Since this date, COVID-19 has spread over the Mexican territory, with over 821,000 accumulated confirmed cases and 83,900 deaths by October 15, 2020 (https://www.gob.mx/salud/documentos/datos-abiertos-152127). Measures for containing virus spreading included a national campaign to promote social distancing, called “Jornada Nacional de Sana Distancia” (National Workday of Healthy Distance) and the closing of non-essential economic activities from March 23 to May 30, 2020. After this lockdown period, gradual reactivation of economic activities was initiated and tuned at the state level using color-coded restriction levels (Acuña-Zegarra, Santana-Cibrian & Velasco-Hernández, 2020). The restrictions are determined taking into account hospital occupancy and its trend, and the incidence rates of each state and its neighbors. Schools, colleges and universities suspended classes and activities are carried out remotely.

An integral comprehension of this epidemic, from different perspectives, is needed to prevent and control it. Presently, a large body of research has been accomplished, principally in the field of medicine. However, it is essential to improve our understanding of how COVID-19 spreads over a territory. For instance, the description of the epidemic characteristics, including spatio-temporal distribution and association with other features, may be useful for identifying covariates, modeling epidemic behavior, and providing reliable information for decision-making. Nevertheless, there are very few studies concerning the spatio-temporal dynamics of COVID-19 in Mexico.

This study aims at investigating the spatial distribution patterns and dynamics of the COVID-19 in Mexico from the beginning of the epidemic, totalizing 6 months by carrying spatial analysis techniques as the computing of spatial autocorrelation and the identification of clustering patterns of the confirmed cases over time.

This stage 1 registered report does not follow a standard report because this research is not based on an experimental design in which independent variables are manipulated and their effect on the dependent variable(s) is evaluated. However, the proposed study is based on robust geo-statistical methods and the statistical significance of the results will be tested. We will analyze the spatio-temporal patterns of the epidemic to test the following hypothesis:COVID-19 pandemic in Mexico is characterized by different clusters evolving in space and time as parallel epidemics.

Connectivity based on a gravitational model allows explaining intermunicipality contagion better than other distances such as Euclidean, least-cost, and resistance distances.

Study area and dataset

The data sets we will use for this study are composed of the epidemiological and geographical auxiliary data:The daily numbers of confirmed COVID-19 cases and deaths for the period from January 1 to October 15, 2020, reported by the Secretary of Health were obtained from the Mexican federal government open data platform (https://www.gob.mx/salud/documentos/datos-abiertos-152127).

Country-based COVID-19 statistics (Total COVID-19 deaths per million people and biweekly cases per million people) from https://ourworldindata.org (University of Oxford).

Digital maps of municipality boundaries, human settlements, and road networks from the National Institute of Geography and Statistics (INEGI).

Population projection produced by the National Population Council (CONAPO) to estimate the current population (CONAPO, 2014).

The records from the Secretary of Health contain additional information such as age, sex, foreign and immigrant status, co-morbidities as well as the state and municipality of residence, and the state of the health unit where consultation occurred. In the present study, focused mainly on spatial patterns, we will aggregate data at the municipal level.

Due to the existence of many asymptomatic cases and the reduced number of diagnostic tests, the epidemiological surveillance of confirmed cases represents only a proportion of all infections. A seroepidemiological survey based on a robust sampling can provide precise estimates of seroprevalence in the population than analysis based on confirmed cases (Pollán et al., 2020). However, Mexico carried out a reduced number of tests and confirmed cases dataset is the only available source at the national level. Limitations of this information are that confirmatory testing is strongly biased towards symptomatic cases, and each state makes its own data collection. However, since the federal government (Secretary of Health) mediates the data aggregation, there is some methodological consistency that allows for homogeneity and reasonable comparisons between states and municipalities. Mexico has had low levels of testing for the virus throughout the entire period. No large variations in the number of cases are expected due to an increase or decrease of the testing efforts.

Since there is a delay in case reporting and confirmation, we will limit this analysis to the period between Jan 1st and October 10, 2020. We also will discard records in which the state of residence was different from the state where the patient did the consultation because these records correspond likely to patients who get infected outside of their municipality of residence. These records represent about 6% of the confirmed cases.

All the analyses will be carried out using the R program (version 4.0.2) (R Development Core Team, 2020), in particular the packages FlexScan 0.2.0 (Tango & Takahashi, 2012), gdistance 1.3-1 (van Etten, 2017), rflexscan 0.3.1 (Otani & Takahashi, 2020), sf 0.9-6 (Pebesma, 2016), and spdep 1.1-5 (Bivand, Pebesma & Gómez-Rubio, 2008).

The geographical database is based on the Lambert conformal conic projection, a conic map projection established on two standard parallels, which minimizes deviation from the unit scale within a region comprising the two standard parallels. Its elaboration will be presented in “Spatio-temporal dataset of COVID-19 outbreak in Mexico” (submitted to Data in Brief; J.F. Mas, 2020, unpublished data). The dataset and the scripts are available at Mendeley Data:Repository name: Data_Mexico_COVID19

Data identification number: 10.17632/mc37xdzw74.1 (DOI)

Direct URL to data: https://data.mendeley.com/datasets/mc37xdzw74/1

Methods

For comparison purposes, we will plot the cumulative deaths per million inhabitants of Brazil, France, Mexico, and the US using the country-based data from the University of Oxford. As the epidemic began at different dates in these countries, the origin (day one) in the time axis will be defined as the day when one death per million inhabitants was reached.

To simplify the data and avoid the day of the week bias, the daily information from the Secretary of Health will be aggregated to the weekly level. The number of new confirmed cases and the number of deaths will be computed for each municipality per week during the period the beginning of community transmission (End of March) and October 10, 2020.

Spatial autocorrelation analysis enables users to test whether the observed value of a nominal or ordinal variable in one place is independent of values of the same variable in neighboring areas (Sokal & Oden, 1978). In this case, the spatial autocorrelation assessment will enable us to evaluate if the municipalities with high or low infection rates tend to be spatially aggregated and form clusters. Moran’s I index is a measure of spatial autocorrelation that can be used to explore the spatial distribution of diseases (Lawson, 2013). It has been used in various studies concerning COVID-19 (Barrantes Sotela & Solano Mayorga, 2020; Cordes & Castro, 2020; Kang et al., 2020; Yang et al., 2020). To depict the spatial association of COVID-19 cases over time, Moran’s I statistic will be computed for each week (Eq. 1).

(1) I=nS0∑i=1n∑j=1nwij(xi−x¯)(xj−x¯)∑i=1n(xi−x¯)2

where xi is the number of cases at location (municipality) i, n the number of municipalities, wij is the weight between observation i and j, and S0 is the sum of all wij’s:

S0=∑i=1n∑j=1nwij

In its simplest form, the weights take values 1 for close neighbors, and 0 otherwise. We will also set wii = 0 because a region cannot be adjacent to itself.

During an epidemic, it is crucial to implement spatio-temporal surveillance that can prioritize locations for specific interventions, rapid tests, and resource allocation. One such method is space-time exploration statistics (Kulldorff, 1997), widely used for different types of diseases (Coleman et al., 2009; Zheng et al., 2014) including COVID-19 (Ballesteros et al., 2020; Desjardins, Hohl & Delmelle, 2020; Hohl et al., 2020). As the following step, spatial scan statistic will be applied to detect and evaluate disease clusters using the “flexibly shaped spatial scan” approach proposed by Tango & Takahashi (2005). This algorithm can detect irregularly shaped clusters such as those along with a linear feature as a road, while algorithms based on a circular window have difficulty in accurately detecting non-circular clusters and tends to define a more extensive cluster than the true one by incorporating surrounding regions (Tango & Takahashi, 2005; Tango & Takahashi, 2012). In epidemic monitoring, the size of a cluster cannot be known a priori, and the population at risk is not evenly distributed. For instance, under the null hypothesis of equal risk of disease inside and outside the cluster, we expect more cases in an urban area compared to a rural area of similar size, due to the higher urban population density. No analytical solutions have been found to obtain the probabilities in these tricky conditions and the algorithm uses Monte Carlo hypothesis test to get the p-values (Kulldorff, 1999).

A large number of candidate clusters is obtained through the creation of irregularly shaped windows on each region (e.g., municipality) by connecting adjacent regions. For each candidate cluster, the number of observed cases will be compared to the number of expected cases of COVID-19, assuming that the COVID-19 cases follow a Poisson distribution. The null hypothesis is that the incidence of COVID-19 is randomly distributed over space, and the alternative hypothesis is that the incidence increases inside the cluster. In order to test whether the clusters are statistically significant, the log likelihood ratio (LLR) will be estimated by Monte Carlo randomization with 999 replications. The p-value is estimated by comparing the rank of the likelihood from the real data set with the likelihood values from the randomized data sets. If this rank is R, then p = R/(1 + Ns), where Ns is the number of simulations. The non overlapping statistically significant clusters will be retained (p ≤ 0.05). Clusters will be estimated for each week, allowing to analyze their temporal evolution by computing the date of their first occurrence and their duration over time.

For each cluster, the relative risk (RR) will be computed (Eq. 2). RR is the risk inside a cluster divided by the risk outside the cluster.

(2) RR=c/e(C−c)/(C−e)

where RR is the relative risk, c is the total number of cases in a cluster, e is the total number of expected cases in a cluster, and C is the total number of cases in the country.

Finally, an attempt will be made to determine which type of distance measure is better suited to explain disease spreading. For that, the distance between municipalities will be calculated using different approaches: (1) Euclidean distance, (2) least-cost distance, (3) resistance distance, and (4) gravity model interaction.

Euclidean distance is a straight line between two locations based only on their coordinates. The other distances, least-cost distance, and resistance distance are based on graph theory. Graphs are obtained by connecting each cell center with its nearest neighbors, which become the nodes of the graph. Weights are associated with each edge and express the conductance (inverse of the resistance). In the present study, the road network map will be rasterized using the speed limit as the value of conductance and a spatial resolution of one kilometer. Cells will be connected with their eight orthogonal and diagonal nearest neighbors (Moore neighborhood). The least-cost distance is the least costly path to travel from one point to another, taking into account the cost associated with moving through space. The distance is expressed in cost units, in this case, time (hours). The resistance distance allows the incorporation of multiple pathways, for instance, the least cost route and alternative ones using secondary roads.

The gravity model enables geographers to model the amount of interaction between two places (Flowerdew & Aitkin, 1982). The expected interaction between two places is proportional to the size of their populations Pi and Pj, and inversely proportional to the square of the distance between them (Eq. 3):

(3) Iij=kPiPjdij2

where Iij is the interaction between places i and j, k is a constant, and Pi, and Pj are the population sizes of places i and j and, dij the distance between these two places.

To determinate the geographical position of each region, instead of using the coordinates of the municipality centroid, the coordinates of the municipality head cities will be used. This decision is based on the fact that, at this stage of the epidemic, most of the confirmed cases are found in largest cities (Villerías Salinas, Nochebuena & Uriostegui Flores, 2020), and the distance between head cities represents the connection between municipalities better than the distance between centroids. Additionally, it is worth noting that the gravitational model will enable us to enhance the relationship between the largest cities.

For each week, the municipality newly incorporated into a cluster will be identified (orange regions in Fig. 1). The first set of distances includes the distances between these municipalities and the municipalities already belonging to a cluster in the previous week (time t − 1). In contrast, the second group comprises distances between the municipalities which remained outside of any cluster (green regions in Fig. 1) and the municipalities already belonging to a cluster (red region). The distances will be calculated using the four approaches described previously. The type of distance that explains the spread of the epidemic better is expected to express a larger closeness between the pre-existing clusters and the newly incorporated municipalities. Therefore, it will increase the difference between the mean distance of the two sets (Eq. 4).

Figure 1 Euclidean Distances between contaminated municipalities a time t − 1 and contaninated and non contaminated municipalities at time t. Other types of distance will be calculated using the same sets of municipalities.

(4) RD=(Dc−Dnc)Dnc

where RD is the relative difference between the two sets of distances, Dc is the mean distance between municipalities that belong to a cluster at weeks t − 1 and municipalities incorporated into clusters at time t. Dnc is the mean distance between municipalities that belong to a cluster a week t − 1 and municipalities that not belong to any cluster at week t.

Additional Information and Declarations

Competing Interests

Author Contributions

Data Availability

The author declares that they have no competing interests.

Jean-François Mas conceived and designed the experiments, performed the experiments, analyzed the data, prepared figures and/or tables, authored or reviewed drafts of the paper, and approved the final draft.

The following information was supplied regarding data availability:

The dataset, including GIS layers, tables, and R scripts, is available at Mendeley Data:

Mas, Jean-François (2020), “Data_Mexico_COVID19”, Mendeley Data, V1, DOI 10.17632/mc37xdzw74.1.

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
