# Peer review of "Stage 1 registered report: spatiotemporal patterns of the COVID-19 epidemic in Mexico at the municipality level"

_PeerJ, doi:10.7717/peerj.10622_

## Round 0.1 · original submission · Major Revisions

Experts in the field have reviewed your registered report manuscript and in agreement with their reviews, I invite you to respond to their comments. Please address their comments line by line making necessary changes.

Reviewer 1 ·

Basic reporting

This manuscript describes the hypothesis and the methods used for making an spatio-temporal analysis of the COVID-19 spread on Mexico.
The manuscript is generally clear, although some corrections should be done in order to reach a higher level of clarity and professional English, specified below.
I find that the introduction lacks some references that may be key for the study. For example, I think that sero-prevalence studies should be cited (eg Pollan et al, The Lancet, doi:10.1016/S0140-6736(20)31483-5), to explain that the statistics typically reflect symptomatic patients, but there may be asymptomatic individuals that are infectious. There are also some studies considering the mobility between municipalities as the connection between them to fit the COVID-19 outbreaks (see Arenas et al, medRxiv, doi:10.1101/2020.03.21.20040022, Eguíluz et al, medRxiv, doi:10.1101/2020.05.03.20089623).

Experimental design

The experimental design is interesting. My main concern is the lack of a link between the clear and well-defined hypothesis and the listed methods, such that the author explains the role of each of the methods. For example, Moran's I index is introduced, but I am missing an explanation of the expected values, what they mean, and what these values will imply for the analysis.

Validity of the findings

For a Stage 1 Registered Report, the description of the hypothesis and the methods is relevant and clear.

Additional comments

I have some additional comments for the author that may help improve the manuscript:
-Abstract: as this is a Stage 1 Registered Report, I think that the abstract should not talk about something done (i.e. in past tense), but about something that will be done.
-L21: I suggest replacing the word region by continent.
-L25-31: which are the sources of this information? Citation needed.
-Additionally to the proposed networks connecting municipalities, is there available census data specifying people that commute (i.e., people that live in one location and work in another)? Two of the proposed references above use similar census data.
-L67-69: more detail is needed. Is this data individual-based, i.e. for each patient you have sex, age... or they are grouped at some aggregation level?
-L79: I think that contaminated is not the best technical word for this context.
-L95-97: first it is proposed to analyse the new number of cases and deaths in Mexico and then it will be compared with the cumulative data in other countries. I propose using the same information for comparing both.
-L138: Euclidean distance is proposed, but the Earth is approximately a sphere. Maybe it would be better to use the Haversine formula for calculating the distances between two points in the surface of a sphere.
-Fig. 1: the color code should be specified, either as a legend or in the caption. The caption states that the figure illustrates the distances, but I just see straight lines, maybe the numbers in km should be annotated next to each line. I also guess this figure does not represent whole Mexico, so 1) an illustration of the data (for example, cumulative cases or deaths) across the whole country would be very illustrative, and 2) it would be good to specify which zone of Mexico is represented in this figure.

Reviewer 2 ·

Basic reporting

Thank you for the opportunity to appraise this registered report which outlines steps that will be taken to identify local coronavirus 2019 (COVID-19) outbreaks in Mexico.
In this manuscript, the authors consider an outbreak detection algorithm–the scan statistic–to examine clusters of COVID-19 in Mexico. The use of scan statistics harmonises well with other approaches found in the literature, particularly papers from the United States of America e.g. Hohl et al. (DOI: 10.1016/j.sste.2020.100354) and Cordes and Castro (DOI: 10.1016/j.sste.2020.100355) who also use Moran’s I in their work.
I commend the authors for following current best scientific practices in submitting their registered research. It is good that authors provide a DOI to a repository where their data is located allowing for the reuse; it is my hope that they will also provide access to a repository with their analysis code. Authors should ensure colours used in figures are colour-blind friendly and are discernable when printed in black-and-white.

Experimental design

The following comments are highest priority and concern the methods used in the approach:
1. (Line 77) My understanding, based on the description provided by the authors, is that the first (imported) case occurred on 28th February 2020. This would mean nearly 1/3 of the data is 0 counts. How does this affect the analysis? (Line 124) Would this mean the assumption of following a Poisson distribution is violated?
2. (Line 92) Aggregating the data to weekly counts gives a time series of 32 observations of which eight-or-so are before the first case occurs. Is this a sufficient sample size for the authors to test their hypothesis of interest?
3. (Line 49) Authors should provide more detail concerning how exactly they intend to test their hypotheses statistically. (Lines 124 and 125) To me it seems that authors are using a Neyman-Pearson decision-making framework rather than Fisher’s significance testing framework. If this is the case, they should specify their sample size, and pre-specified limits for type I and type II errors. (Line 126) Which values of the likelihood ratio would be considered significant?
4. (Line 82) For reproducibility purposes, authors should note which versions of software and software packages they use
5. (Lines 46-50) Authors should be aware that reporting standards do exist for registered reports that are not based on experiments. Authors should look at e.g. van den Akker et al. (DOI: 10.31234/osf.io/hvfmr)
6. a. (Line 64) It is unclear to me why authors are using population projections rather than census data to obtain population counts. The Mexican national institute of statistics, geography, and information (INEGI) should have been conducting a nationwide population census in March of this year (censo2020.mx) so updated population counts should be available. b. Authors could consider using the United Nations World Population Projections as a sensitivity analysis, as these are constructed under a couple of different scenarios. However the UN projections may only exist at national level rather that subregional
7. (Line 71) Since the analysis period covers a period of time when testing efforts may have been increased, this paragraph and its effect on the analysis should be discussed in more detail
8. (Line 157) These considerations concerning large cities are captured in the gravity model approach (DOI: 10.1086/422341) and authors may wish to note that more explicitly
The following comments will hopefully provide more detail for the reader to understand the situation being studied:
1. (Line 21) In the introduction I would suggest authors provide exact number of cases to support their point that the Americas are the most affected regions. If such information is sourced from the World Health Organization (WHO), I would further suggest authors use WHO regional terminology, which in this case the region would be WHO PAHO
2. (Line 30-31) At the end of the second paragraph in the introduction it would be nice to have some more information about the situation in June, July, and August as the analysis also spans those months and the only policy information provided for the reader to understand the context concerns March to May
3. (Line 61) Authors should note exactly which information they are using from the “Our World in Data” initiative to allow readers to access the same information, “country-based COVID-19 statistics” is vague
4. (Line 138) Authors should motivate their choice of distances considered, it seems there may have been an additional option which was since removed

Validity of the findings

no comment - section will become relevant after IPA

Additional comments

The following comments are suggested minor changes to text:
1. (Line 51) “COVID” should be “COVID-19”
2. (Line 29) Anda-Jauregui used the term “Journada Nacional de Sana Distancia” (arXiv: 2006.11635) in their manuscript. I am not a Spanish speaker so I am not sure whether this is the correct term and “Sana distancia” is more colloquial
3. (Line 107) Authors should note what n and x’s are
4. (Line 109) Authors should explain that w_{ii} = 0 means a region cannot be adjacent to itself
5. (Line 138) Authors should update numbering such that resistance distance is 3 and gravity model is 4
6. (Line 152) Uppercase “K” should be lowercase “k” to match notation used in formula above
7. (Line 152) Authors should note what d_{ij}^2 is
8. (Lines 132, 152, and 169) Authors might wish to note why denominators cannot be zero
9. (Line 172) Time is usually denoted t rather than w. This may avoid confusion with weights that are also denoted w

---

## Round 0.2 · accepted · Accept

Thank you for addressing the reviewer comments. Reviewer 1 had a couple of minor comments that I think may improve your future results publication and hence I can go ahead with my decision to accept your manuscript. Best wishes in your research.

Reviewer 1 ·

Basic reporting

Thanks to the author for addressing our comments.
There are still two minor changes that I suggest:
-I proposed to provide examples of limit cases of Moran's index and explain what they mean, which could improve very much the understanding of the reader.
-Euclidean is not spelled right in the caption of Figure 1.
Probably the second point can be modified at the proof stage, and the first is just a remark, as I already mentioned that, but it seems that the author decided not to include it.

Experimental design

No new comments.

Validity of the findings

No new comments.

Reviewer 2 ·

Basic reporting

I believe the authors have addressed the points I raised in the previous review in a satisfactory manner and I have no further comments at this time. I look forward to seeing the results once the study has been conducted.

Experimental design

See previous comment

Validity of the findings

NA until after analysis conducted